# Enhancement of Skill Competencies in Operative Dentistry Using Procedure-Specific Educational Videos (E-Learning Tools) Post-COVID-19 Era—A Randomized Controlled Trial

**DOI:** 10.3390/ijerph19074135

**Published:** 2022-03-31

**Authors:** Azhar Iqbal, Kiran Kumar Ganji, Osama Khattak, Deepti Shrivastava, Kumar Chandan Srivastava, Bilal Arjumand, Thani AlSharari, Ali Mosfer A Alqahtani, May Othman Hamza, Ahmed Abu El Gasim AbdelrahmanDafaalla

**Affiliations:** 1Department of Operative Dentistry & Endodontics, College of Dentistry, Jouf University, Sakaka 72345, Saudi Arabia; dr.azhar.iqbal@jodent.org; dr.osama.khattak@jodent.org; 2Department of Operative Dentistry & Endodontics, Frontier Medical and Dental College, Abbottabad 22010, Pakistan; 3Department of Preventive Dentistry, College of Dentistry, Jouf University, Sakaka 72345, Saudi Arabia; sdeepti20@gmail.com; 4Department of Periodontics, Sharad Pawar Dental College & Hospital, Datta Meghe Insitute of Medical Sciences, Nagpur 440016, India; 5Department of Oral & Maxillofacial Surgery and Diagnostic Sciences, College of Dentistry, Jouf University, Sakaka 72345, Saudi Arabia; drkcs.omr@gmail.com; 6Department of Conservative Dental Sciences and Endodontics, College of Dentistry, Qassim University, Qassim 52571, Saudi Arabia; ba.ahmad@qu.edu.sa; 7Restorative and Dental Materials Department, Faculty of Dentistry, Taif University, Taif 26571, Saudi Arabia; thani.alsharari@gmail.com; 8Department of Diagnostic Dental Sciences, College of Dentistry, King Khalid University, Abha 62529, Saudi Arabia; al.alqahtani@kku.edu.sa; 9Department of Prosthetic Dental Sciences, College of Dentistry, Jouf University, Sakaka 72345, Saudi Arabia; dr.may.hamza@jodent.org; 10Department of Urology, King Abdulaziz Specialist Hospital, Sakaka 72345, Saudi Arabia; yazangasim@gmail.com

**Keywords:** dental skills, operative dentistry, procedure-specific videos, E-learning, dental education

## Abstract

E-learning has completely transformed how people teach and learn, particularly in the last three pandemic years. This study evaluated the effectiveness of additional procedure-specific video demonstrations through E-learning in improving the knowledge and practical preclinical skills acquisition of undergraduate dental students in comparison with live demonstration only. A randomized controlled trial was conducted for the second-year dental students in the College of Dentistry, Jouf University, to evaluate the impact of E-learning-assisted videos on preclinical skill competency levels in operative dentistry. After a brief introduction to this study, the second-year male and female students voluntarily participated in the survey through an official college email. Fifty participants were enrolled in the study after obtaining informed consent. The participants were randomly divided into two groups, twenty-five each. The control group (Group A) was taught using traditional methods, and the intervention group (Group B) used E-learning-assisted educational videos and traditional techniques. An objective structured practical examination (OSPE) was used to assess both groups. The faculty members prepared a structured, standardized form to evaluate students. After OSPE, statistical analysis was done to compare the grades of OSPE between Group A and Group B. Logistic regression analysis was done to express the effect of components of the OSPE on gender, cumulative gross point average (CGPA), Group A and Group B. The results showed a significant difference in the experimental groups after the intervention (*p* < 0.000). The simulator position parameter demonstrated that the participants had a significant competence level after the intervention by procedure-specific videos (*p* < 0.000) and an exponential value of 6.494. The participants taught by E-learning-assisted procedure-specific videos and traditional teaching strategies demonstrated an enhanced learning and skill competency level than participants who used only traditional teaching strategies.

## 1. Introduction

The undergraduate dental program concentrates on the students’ psychomotor skill development throughout their early preclinical years, often notably applicable to the preclinical operative courses in the dentistry curriculum. The students are exposed to the technical aspects of practical preclinical skills, which play a vital role in boosting the outcome of clinical procedures [1]. Usually, the dental faculties use a standard, traditional live demonstration methodology for teaching preclinical laboratory skills. The live demonstration in small clusters has been helpful in teaching the preclinical laboratory skills due to its improved communication skills and accumulated student confidence, and provided a higher understanding of the procedure than informative, didactic teaching [2,3]. However, various studies have shown that the live demonstration–based teaching methodology has several drawbacks, such as difficulty in the visualization of the procedure, the reliance on the students on the instructor, and slight variations of the procedure among the different instructors [2,3,4]. Aside from these drawbacks, the live demonstrations’ effectiveness depends on the number of students allotted to the instructor and the amount of time spent delivering the live demonstration. Another issue associated with the live demonstration is, it is delivered only once for a particular selected procedure. Therefore, students might not get an opportunity to repeatedly follow the procedure to understand and master the essential skills. According to one study, the traditional teaching methodology has caused significant psychological distress, which results in anxiety, depression and burnout among the students [2]. Due to the present situation of the COVID-19 pandemic, the students and faculty had to follow social distancing norms that made it even worse for the students to visualize the procedure properly. Keeping in view these drawbacks of traditional teaching methodologies, and conjointly because of the technological advancement in the past few decades [5,6,7,8], it has become essential for educators and clinicians to bring their teaching styles and methodology in line with the current pandemic situation and the students’ learning needs and training desires to reinforce and improve their knowledge and preclinical and clinical skill competency [9,10,11,12]. There is a place for vicarious and experiential learning strategies in clinical skills training. Clinical teachers must use learner-centered ways to get to know their students, as well as their students’ strengths, limitations, talents and experiences [13]. Therefore, educators and clinicians are trying to find new teaching methodologies for preclinical laboratory skills. Procedure-specific educational videos and video demonstrations could be blended and integrated with preclinical live demonstrations. These procedure-specific educational videos permit students to visualize the procedural steps within the lab on the projector and E-learning tools on Blackboard, on-campus and off-campus [14]. This will allow the students to revise the procedural steps before, during, and after the skill lab session as per the students’ convenience [15]. It also reduces information differences and bias and provides uniformity in learning experiences for all students [2]. Since we have entered a digital era, the concept of E-learning in various forms has emerged as an effective tool for teaching strategies [16]. E-learning can be defined as learning while utilizing electronic technologies to access educational curricula outside a traditional classroom [17,18]. Video demonstration through one of the E-learning tools, such as Blackboard has been witnessed as an implicit tool. Literature suggests that students are more inclined toward the newer teaching method than traditional learning [19]. Recently, Elham Soltanimehr discovered that virtual learning is better than traditional lecture-based learning for knowledge acquisition augmentation during the diagnostic imaging of bone lesions of the jaw [20]. Therefore, he has suggested that virtual educational programs must be revised to improve the student’s reporting skills [20]. Kon H et al. showed in their study that the videotapes were considered valuable resources because a better visualization of the procedure can be achieved by repeated replay and review functions. Hence, videotapes were considered a useful and valuable recap tool for the clinical demonstration during denture construction [21]. Wong et al., in their study, showed that the use of instructional videos has been found effective in complementing the advanced trauma life support approach for teaching psychomotor skills in the administration of local anesthetics by oral health students [19]. Due to the present pandemic situation of COVID-19 and the lack of or limited in-campus teaching due to social distancing restrictions, we were compelled to enhance students’ knowledge and skills by utilizing the support of E-learning. This study hypothesized whether the addition of procedure-specific video demonstrations (E-learning) improves the acquisition of knowledge and preclinical practical skills for undergraduate dental students in comparison with live demonstrations only.

## 2. Methods

Male and female second-year dental students from the College of Dentistry, Jouf University, who passed their prerequisite courses for the preclinical operative dentistry skill course, were recruited for the study voluntarily after signing an informed consent form. The dental undergraduate students with any psychomotor disability were excluded from the study. The study was conducted during the scheduled laboratory hours in the college premises after ethical approval was taken from the local bioethics committee with no 254-1-2020, as per institutional policy. Using computer-generated random numbers, these participants were randomly divided into the control group (Group A) and the experimental group (Group B), using simple randomization sampling based on teaching methodology (*n* = 25). Sample size calculation was done by using the G*Power computing tool (Heinrich-Heine-Universität Düsseldorf, Germany), a confidence interval (α) of 0.05 and power (1 − β) 95–0.95%, the difference between the two dependent means (matched pairs) and effect size (f) of0.5. The total sample size generated was found to be five; however, four students were unable to participate in the study due to some personal reasons. The skill-based procedure under evaluation was a part of the preclinical second-year dental undergraduate curriculum.

The participants in the control group (Group A) were taught about the skill-based procedure with the routine live lab demonstration. Contrarily, the participants in the experimental group (Group B) were taught by using a procedure-specific educational video demonstration through an E-learning tool (Blackboard) plus the routine live lab demonstration, described as a hybrid (Figure 1). The live demonstration and procedure-specific educational video described the identical steps for the class I cavity preparation and amalgam restoration on typodont tooth no. 36. The procedure involved 330 carbide burs in a high-speed handpiece with air–water spray, a mouth mirror, explorer and periodontal probe. The live demonstration was given by an experienced academician who handled preclinical and clinical work. The procedure-specific educational video was also produced by the same faculty member who gave the live demonstration to avoid any information differences regarding the procedure.

On the day of the live demonstration, the procedure-specific video was sent through Blackboard to the experimental group (Group B) participants who were instructed by a Blackboard announcement and official institutional email to watch the procedural-specific videos before attending the evaluation session for the same procedure in the next scheduled lab hours for both groups. A request was made to the institutional E-learning unit that access should not be provided to the control group (Group A) participants. The video could not be recorded or copied by the experimental group (Group B) participants to avoid sharing it with the control group (Group A) participants. Statistical tracking through Blackboard was enabled to ensure that participants viewed the video for a minimum of five to six views. After one week, an objective structured practical examination (OSPE) was conducted for both groups to assess the clinical competency level of achievement resulting from this intervention. The OSPE consisted of six stations: infection control and operator position, tray organization, simulator position, cavity outline and extension, resistance form and retention form, which were considered outcome variables. To maintain the inter-examiner reliability, Cohen’s kappa statistic was used to measure the agreement level between the scores of the OSPE assessed by two faculty members who were not involved in any of the steps of this study. The kappa score between the two faculty members was found to be 0.88. The participants of both groups were anonymized by the examiners. The average score of the two examiners was taken as the score of that OSPE station. The final average scores of all the OSPE stations were calculated, but these did not contribute to the student’s midterm or final term exam grades. It was made sure by the course organizer that the examiners did not share the result. At the end of the study, a questionnaire containing seven questions was circulated among the students to get their feedback, assess their perception of the teaching methodologies, and compare them.

### Statistical Analyses

After any necessary editing, the biographic and assessment sheet data were transferred into an Excel sheet. The participant’s personal information was treated anonymously for their privacy and confidentiality; therefore, a code/sequence was given to each subject for identification. The descriptive analysis (presentation of data in the form of percentage and mean with standard deviation) and inferential analysis, such as the McNemar test were used to assess the test of significance related to competent and non-competent between the control and experimental groups, and logistic regression analysis was done to express the effect of OSPE performances on gender, CGPA, control and experimental group, using a statistical package for the social sciences (SPSS IBM, version 21, Chicago, IL, USA).

## 3. Results

There were 50 participants, 28 males (56%) and 22 females (44%), in this study. These participants were randomly divided into two groups: control and experimental. There were 25 participants in each group (*n* = 25), 14 males (56%) and 11 females (44%). The results (Table 1) show that there was no significant difference before the intervention between the control group and the experimental group in the competence level of the participants, but a significant difference was found in the experimental groups (Table 2) after the intervention. The *p* value was 0.000. Among all the study variables, only a simulator position has shown that the participants had a significant competence level after the intervention by procedure-specific videos, with a *p* value of 0.000 and an exponential value of 6.494.

Binomial logistic regression was performed to ascertain the effects of infection control, operator position, tray organization, simulator position, cavity outline and extension, resistance form and retention form parameters on the likelihood that participants were successful in the skilled procedure. The percentage of variance in being competent within the measured parameters, such as for infection control was 52.3%, operator position 54%, tray organization 60%, simulator position 45%, cavity outline and extension 61%, resistance form 65%, and retention form 65 %.

The effect of the infection control parameter on the students’ competence level demonstrated a significant difference between males and females (*p* value, 0.000; odds ratio, 40.3) in which females were considered constant. Under the CGPA group, there was a significant difference with different levels of CGPA score (*p* value, 0.000; odds ratio, 325). Finally, concerning infection control, there was a significant difference among groups 1, 2, and 3 (*p* value, 0.000). The effect of operator position on the students’ competency level demonstrated a significant difference between male and female groups (*p* value, 0.001; odds ratio, 40.389). Regarding the CGPA group, there was a significant difference with different levels of CGPA score (*p* value, 0.000; odds ratio, 325.42). There was no significant difference found among groups 1, 2, and 3 regarding the operator position. Regarding the tray organization, a significant difference was found regarding CGPA level (*p* value, 0.003; odds ratio, 25.916) according to the binary logistic regression analysis results. Regarding the genders and among different groups, no significant difference was found. Regarding the cavity preparation outline and extension, a significant difference was found (*p* value, 0.004; odds ratio, 0.044) in group 3 among the different groups. A significant difference was found (*p* value, 0.003; odds ratio, 51.010) among the different CGPA levels during the binary logistic regression analysis (Table 3, Table 4, Table 5, Table 6 and Table 7).

The logistic regression model was statistically significant concerning the simulator position parameter, which influences the students’ competence level (Table 4). The model’s sensitivity was 0% with 100% specificity, the positive predictive value cannot be calculated, and the negative predictive value was 65%. The comparison of the participant’s perceptions about the two teaching methodologies showed that procedure-specific videos through E-learning helped the participants in the repetition of skills (4.70 ± 0.398), can be used as an adjunct teaching tool (4.26 ± 0.395), made the participants feel more competent in performing the skill-based procedure (4.30 ± 0.564), and helped the participants to better understand the preclinical practical lab skills (4.20 ± 0.538).

## 4. Discussion

The dentistry curriculum emphasizes developing psychomotor skills to effectively and judiciously treat patients [18]. The psychomotor skills training for dental undergraduates in operative dentistry starts in the preclinical laboratory. These preclinical laboratories are the foundation stones for inculcating the expertise required to treat patients in clinics during their clinical curriculum. Traditionally, preclinical procedures are taught with the help of live demonstrations in preclinical laboratories [21,22]. This study has found that the students who were given procedure-specific video demonstrations were more competent in preclinical skills than students taught in traditional learning. Khalaf K et al., in their study, concluded that video-assisted learning as an additional tool to traditional teaching could augment the understanding and learning process of students [17]. Thilakumara IP et al. have found in their study that there was a statistically significant difference in terms of improvement of knowledge in the group that used the procedural video [18]. Fayaz A et al., in their study, concluded that instructional videotapes could aid in teaching the fabrication of complete dentures and are as effective as the traditional teaching system [23]. Recently, Elham Soltanimehr has documented that the virtual method of learning was better for acquiring knowledge than the traditional lecture-based learning during the diagnostic imaging of bone lesions of the jaw [20].

The competency level showed no significant difference between the control group and experimental group before the intervention, indicating the same level of knowledge and competency of dental undergraduates participating in the study. After the intervention, the experimental group in which procedure-specific video demonstration was given, has shown a significant difference. Similar results are obtained in other studies [24]. These differences in the competency level before and after the intervention can be contributed by the fact that video demonstrations enable the student to visualize the procedure [18]. At the same time, it might help them recall the process and implement it in preclinical activities [18]. The procedure-specific educational videos are vital because they allow the students visual and mental practice and enhance their psychomotor skills during the procedures and the novel aspects of learning from videos [25]. Moreover, it is beneficial to acquire technical skills and simulation in clinical settings [20,21]. However, contrarily, some studies have found no difference in the competency level of the students, whether they have been given a video demonstration or traditional teaching [26]. Because the students have different psychomotor skill levels, dividing them into other groups and evaluating the effect of the two methods might not necessarily show the real impact [26].

In the preclinical skill of operative dentistry, cavity preparation and restoration are the main procedures. They involve multiple steps, which second-year undergraduate dental students should master. These steps may range from general steps, including infection control, simulator and operator position, and tray organization, to the specific steps of cavity preparation, including cavity outline, resistance form and retention form. Therefore, students’ understanding and mastering of these steps were crucial and were evaluated using two different teaching methodologies by two other students in this study. When infection control and operator position were compared, male students were 1.60 times at higher risk of noncompliance than females. Contrarily, male students had shown an 81% higher chance of higher competency when the simulator position was compared to female students. Contrarily, some studies have reported no gender variation related to competency level in preclinical prosthodontics laboratory techniques [18].

Similarly, students with low CGPA scores were 2.51 times at higher risk for not showing competence in infection control procedures, 1.41 times at higher risk for not properly arranging the tray, and 2.51 times at higher risk of using incorrect operator position. On specific cavity design, students with low CGPA were 1.70 times at higher risk of competency than high CGPA scores in the resistance form and retention form of cavity preparation. The effect of CGPA could be related to the confidence level of the students. It has been reported that students with high CGPA scores were able to perform specific dental procedures better than the students with low CGPA scores [27].

There has been a considerable debate on implementing newer teaching strategies over the past years. However, in E-learning, the learner may take a more self-directed learning approach. Nevertheless, self-directed learning is one of the essential adult learning methods that can prepare dental students for a successful lifelong career as a dentist [28]. Regardless, if the system is introduced strategically with proper planning, it may influence the students’ quality of learning [29]. The procedural video demonstration method helps students to gain knowledge and visualize the steps.

Furthermore, it is a self-paced method, and students are given a chance to watch and understand the procedure by watching the videos multiple times at their convenience. It helps improve levels of motivation, satisfaction and concentration [30]. While in traditional teaching, the instructor gives a live demonstration in a shorter period, and sometimes, a few technical steps are difficult to visualize from one direction. A student may be allowed to repeat the procedure themselves (experiential learning) after a live demonstration. The demonstrator will then assess the student’s work and provide constructive feedback [13]. It allows dental students to plan productively for their next learning experience, thereby enabling progress around the experiential (learning by doing) learning cycle. Besides supporting reflection, this feedback also helps students gain a more in-depth understanding of complex subjects.

Moreover, sometimes, there are variations seen in the live demonstrations of different instructors [2,3,4,5]. A simulated dental environment is often used for live demonstrations, which is crucial for dental students’ familiarity and community of practice. Even so, E-learning may occur in environments other than the dental environment.

While assessing the participant’s perception of the teaching methodologies to compare them at the end of this study, we have found that the procedure-specific videos through Blackboard were considered a better teaching methodology than the live demonstration. Therefore, procedure-specific video demonstration can be an alternative method for live demonstrations on five out of seven statements. The finding agreed with the study done by Alqahtani et al., which showed a high mean response for the procedure-specific video group (experimental group) than the live demonstration group (control group) concerning understanding the different steps, visualization, and clarity of the procedure [3]. In a study by Argon and Zibrowski, the participants preferred procedure-specific videos over the live demonstration, claiming that they were able to visualize better and had the liberty to review the procedure at any time as per their convenience and as many times as required [31]. Although every effort was made to standardize the procedure, individual variations of the lab instructor can affect the live demonstration even then.

Although there is a shred of evidence that procedure-specific video-based teaching and learning are preferred methods by students, certain studies show contrasting findings of the participants’ attitudes toward procedure-specific video-based learning. A study by Smith et al. found no difference between the attitude of medical students toward the method of teaching and instruction [32]. A novelty of this study is that results were obtained from a comparative group and a control group via convenient sampling. As a control group, it was chosen because small cohorts of students could share resources and learn together.

Many variables can influence the students’ competence level in learning practical skills. Educational videos can only be used as adjunct tools, not as alternative tools for the learning process. E-learning tools’ most significant limitation is the lack of demonstrator interaction, whereas, in live demonstrations, the demonstrator can clarify students’ questions during the demonstration process. Contrarily, the live demonstration could promote the social learning theory of community of practice (students and faculty members are part of a group who share a common interest and a desire to gain knowledge from and contribute to the community with their variety of experiences) [33,34]. Learning by educational videos does not support the concept of directed self-learning. Furthermore, the abrupt online transition of the learning process during COVID-19 can negatively affect legitimacy and validity. Other factors include a lack of practical skills, low attendance due to heavy internet traffic, and student involvement. Despite the contribution of this study to the literature, there are a few unanswered questions. Since this study was conducted on a single cohort of samples from one speciality of dentistry, it is not easy to generalize its results to other branches of dentistry, such as periodontics or prosthodontics. This study shows the longitudinal impact of educational videos on knowledge and skills retention and how it is transferred into safe clinical practice. Research in the future should investigate students’ levels of competency in restoring teeth after watching the supplemental videos in clinical practice sessions.

## 5. Conclusions

The participants taught by hybrid teaching modality proved to be better and demonstrated a higher level of knowledge and skill competency than those who were not. Therefore, we recommend within the scope of this study that additional procedure-specific educational videos and other resources through E-learning should be a part of the teaching methodology for the preclinical operative dentistry skill course to enhance the students’ knowledge and skill competency levels.

## Figures and Tables

**Figure 1 ijerph-19-04135-f001:**
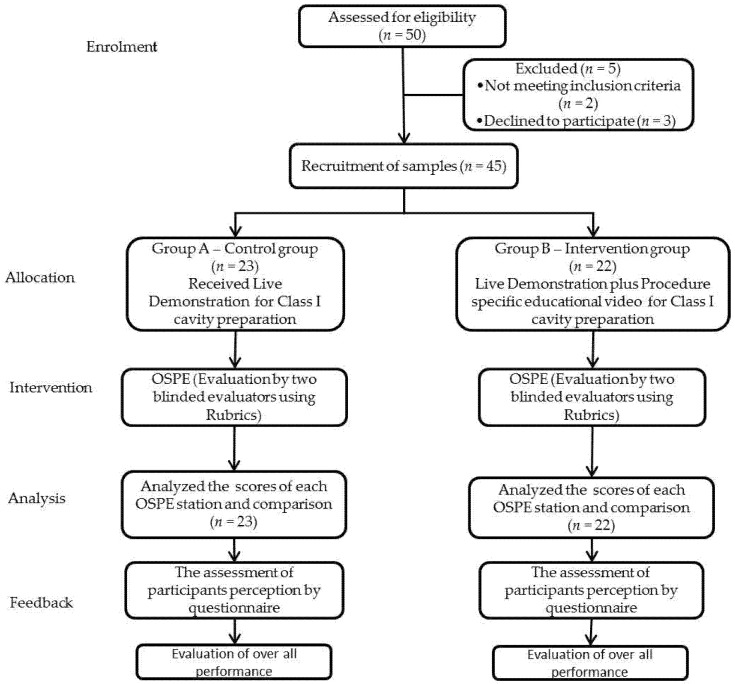
Summary of research methodology and experimental protocol.

**Table 1 ijerph-19-04135-t001:** McNemar test analysis for competent and non-competent in the control group.

2 × 2 Contingency Table for Control Group (Before & After)					
		Control Group After		Total	*p*-Value
		Non-Competent	Competent		
Control group before	Non-competent	122	0	122	0.352
	Competent	12	41	53	
Total		134	41	175	

**Table 2 ijerph-19-04135-t002:** McNemar test analysis for competent and non-competent in the experimental group.

2 × 2 Contingency Table for Experimental Group (Before & After)					
		Experimental Group After		Total	*p*-Value
		Non-Competent	Competent		
Experimental group before	Non-competent	34	0	34	0.000
	Competent	22	119	141	
Total		56	119	175	

**Table 3 ijerph-19-04135-t003:** Expressing the effect of infection control and operator position parameters on gender, CGPA, Group A and Group B using binary logistic regression analysis.

Parameter	Variables	B	S.E.	Wald	df	Sig.	Exp (B)	95% C.I. for EXP (B)	
								Lower	Upper
Infection control & Operator position	Gender	3.69	1.10	11.17	1	0.001	4.38	4.62	5.097
	CGPA	5.7	1.47	15.448	1	0.000	5.25	18.17	5.25
	Group A	0.66	0.82	0.644	1	0.422	1.93	0.38	9.68
	Group B	1.2	0.72	2.78	1	0.095	3.32	0.81	13.65
	Constant	−26.99	6.94	15.11	1	0.000	0.000		

**Table 4 ijerph-19-04135-t004:** Expressing the effect of tray organization parameter on gender, CGPA, Group A and Group B using binary logistic regression analysis.

Parameter	Variables	B	S.E.	Wald	df	Sig.	Exp (B)	95% C.I. for EXP (B)	
								Lower	Upper
Tray Organization	Gender	0.80	1.05	0.58	1	0.445	2.23	0.28	3.68
	CGPA	3.25	1.10	8.72	1	0.003	25.91	2.98	3.45
	Group A	−1.33	0.84	2.47	1	0.115	0.26	0.05	1.38
	Group B	1.08	0.88	1.48	1	0.223	2.94	0.51	2.76
	Constant	−13.14	5.24	6.28	1	0.012	0.00		

**Table 5 ijerph-19-04135-t005:** Expressing the effect of simulator position parameter on gender, CGPA, Group A and Group B using binary logistic regression analysis.

Parameter	Variables	B	S.E.	Wald	df	Sig.	Exp (B)	95% C.I. for EXP (B)	
								Lower	Upper
Simulator position	Gender	1.87	0.92	4.13	1	0.042	6.49	1.06	6.46
	CGPA	2.41	0.89	7.20	1	0.007	11.17	1.91	65.11
	Group A	−1.68	0.75	5.04	1	0.025	0.18	0.04	0.80
	Group B	0.00	0.69	0.00	1	1.000	1.00	0.25	3.92
	Constant	−10.15	4.24	5.71	1	0.017	0.00		

**Table 6 ijerph-19-04135-t006:** Expressing the effect of cavity outline and extension parameter on gender, CGPA, Group A and Group B using binary logistic regression analysis.

Parameter	Variables	B	S.E.	Wald	df	Sig.	Exp (B)	95% C.I. for EXP (B)	
								Lower	Upper
Cavity outline and extension	Gender	−1.65	1.18	1.93	1	0.164	0.19	0.01	1.96
	CGPA	1.72	0.99	3.04	1	0.081	5.62	0.80	3.14
	Group A	−2.15	1.02	4.40	1	0.036	0.11	0.01	0.86
	Group B	0.33	0.82	0.16	1	0.684	1.39	0.28	6.97
	Constant	−4.60	4.89	0.88	1	0.347	0.01		

**Table 7 ijerph-19-04135-t007:** Expressing the effect of resistance form and retention form parameter on gender, CGPA, Group A and Group B using binary logistic regression analysis.

Parameter	Variables	B	S.E.	Wald	df	Sig.	Exp (B)	95% C.I. for EXP (B)	
								Lower	Upper
Resistance form & Retention form	Gender	0.04	1.09	0.00	1	0.965	1.04	0.12	8.89
	CGPA	3.93	1.31	8.98	1	0.003	11.01	3.89	5.36
	Group A	−0.23	0.93	0.06	1	0.804	0.79	0.126	4.99
	Group B	2.09	0.95	4.83	1	0.028	8.09	1.25	3.57
	Constant	−16.28	6.20	6.88	1	0.009	0.00		

## Data Availability

Data will be made available upon request.

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
