# Peer review of "Enhancement of Skill Competencies in Operative Dentistry Using Procedure-Specific Educational Videos (E-Learning Tools) Post-COVID-19 Era—A Randomized Controlled Trial"

_ijerph, 2022, doi:10.3390/ijerph19074135_

Round 1

Reviewer 1 Report

Manuscript ID: ijerph-1607432

Enhancement of skill competencies in Operative Dentistry using Procedure Specific Educational Videos (E-learning tools) post COVID 19 era - A randomized controlled trial

Thank you for giving me the opportunity to review this manuscript. There are many issues with this manuscript. It is not very well written, which makes difficult to follow or understands the authors intend in many areas. I also suffer from typos and sentences structure issues. Therefore, I suggest that it is proof read well before any potential resubmission.

Abstract:

Line 31, it reads: “All fifty male and female students were agreed to participate in the study.”

Comment:  Correct this sentence.

Line 40 and 41, it reads: “. Simulator position demonstrated that the participants had a significant competence level after the intervention by procedure-specific videos. etc.”

Comment: What is Simulator position demonstrated mean? It is not clear in the abstract what this means.

Please correct accordingly.

Introduction:

This section need to be rewritten and proof read before any potential resubmission. It suffers from typo and sentence structuring issues. For example on line 67 , 68 and 69, it reads; “Due to the present situation of COVID 19 pandemic, the students and faculuties had to follow the social distancing norms that made it even worse for the students to visualize the procedure properly.”

Comment: see underlined and bold.

Methods and Materials:

I believe this section should be titled as method since there are no materials involved.

In lines 108 to 110, it reads, “Male and female second-year dental students of College of Dentistry, Jouf University, who have passed their prerequisite courses for the pre-clinical operative dentistry skill course were recruited in the study voluntarily after signing an informed consent form.

Comment: What is the study designed to investigate if the participants have already “passed their prerequisite courses for the pre-clinical operative dentistry skill course”?

Please clarify.

In lines 11 and 112, it reads, “The dental undergraduate students with any psychomotor disability were excluded from 111 the study.”

Comment: The term psychomotor disability is an unusual term. If this term is to be accepted, how can there be students in dentistry categorized, according to the authors, as having a psychomotor disability?

Please clarify.

In lines 131, 132 and 133, it reads, “The live demonstration was given by a faculty member who has completed his fellowship in operative dentistry and has more than eight years of academic experience handling pre-clinical and clinical work.

Comment: What does it mean “completed his fellowship in operative dentistry”. It is not clear what this fellowship is. Perhaps it is sufficient to say an experienced academician who has been involved in handling pre-clinical and clinical work.

In lines 144 and 145, it reads, “Statistical tracking through Blackboard was enabled to ensure that participants have viewed the video for a minimum of 5 to 6 views.

Comments: Group B (test) viewed the procedure 5 to 6 times, and Group A viewed the live demonstration only once. Apart from being unfair to group A, this would skew the results in favor of the video live demonstration. 

It reads on lines 146 to 150 “To maintain the inter-examiner reliability, two qualified faculty members, who were contributors in the same pre-clinical course but did not involve in any of the steps of the present study, were invited by the course organizer to assess the OSPE of each participant as examiners by using five-point rubrics for the procedure.”

Comment: This is not a reliable method of conducting inter-examiner reliability since there could still be a difference in evaluation between the two examiners. The way to do that and the use of Kappa analysis is well documented in the literature.

In lines 154 to 157, it read, “At the end of the study, a questionnaire containing seven questions was circulated among the students to get their feedback to assess their perception about both the teaching methodologies and compare them.

Comment: there are many issues with stating a questionnaire was used or circulated. First, what is the validity and reliability of this questionnaire? Or was it a survey rather than a questionnaire? Secondly, in what language was it made? If any in Arabic, and given that the reporting of the data is in English, was linguistic validity made? 

The methodology needs to be rewritten with the points above taken into consideration. There is an essential point concerning the methodology; it seems Group A (control group) had the live demonstration, while group B had both live and void demonstration, which should be described as “hybrid.”

Of significance, is who was involved in the recruitment of students’ participants? The students should not be recruited by the course director, as they may feel under pressure to participate in the study.

Results:

It reads in line 174 to 177 “The results (Table 1) shows that there was no significant difference between the control group and the experimental group before the intervention in the competence level of the participants, but a significant difference was found in the experimental groups (Table 2) after the intervention and the p-value was 0.000.

Comment: How was the difference between the control and experimental groups before the intervention in the competence level evaluated?

Please clarify

Discussion:

The discussion needs to be re-written, focusing on the results outcomes.

This study does not show, describe or discuss limitations, and any research study would have some degree of limitation. In this study, there are several limitations, but the authors have not addressed them. For example, the fact that the study was a single-center and done among one group/year would be a limitation since generalizability cannot be made.

End report

Author Response

We thank the reviewer for his valuable inputs on our manuscript. It was a pleasure to respond to the respected reviewer's comments. The constructive comments raised the standard of the manuscript and increased its international appeal. All the comments raised by the reviewer are being addressed in point to point clarification file which is attached herewith. 

Reviewer 2 Report

  1. Please consider the addition of: dental education to the keywords list.
  2. A pertinent point to add is that the YouTube platform, possibly others to include social media outlets, now contain a huge range of dental procedural videos. Therefore, dental students of today can also access these to supplement their educational institute’s e-learning/Blackboard. Combined with live demonstrations and practical experience, a lot of dental students may be much more privileged than past generations.
  3. Learning through watching others (i.e., faculty members, dentists, allied professionals etc) or vicarious learning is an important learning approach in operative dentistry training. This should be discussed in the Introduction section. A relevant article that will be a useful read, and that can be referenced with regards to this: Modha, B., 2021. Experiential learning without prior vicarious learning: An insight from the primary dental care setting. Education for Primary Care32(1), pp.49-55.
  4. The article seems very positive and championing about e-learning. However, there needs to be more of a balance to include the drawbacks. A relevant article that will be a useful read, and that can be referenced with regards to this: Shersad, F,. & Salam, S,. (2020). Managing Risks of E-learning During COVID-19. DOI: 10.13140/RG.2.2.12722.63689 (https://www.dmcg.edu/wp-content/uploads/2020/10/IJIRES_1760.pdf)
  5. Live demonstrations provide face-to-face contact. This may allow students to ask their demonstrators questions to have their queries addressed – this could promote the social learning theory of community of practice (students and faculty members are part of a group who share a common interest and a desire to learn from and contribute to the community with their variety of experiences), and this may supplement their learning (reference: Abidi, S.S.R., 2007. Healthcare knowledge sharing: purpose, practices, and prospects. In Healthcare Knowledge Management(pp. 67-86). Springer, New York, NY). E-learning may not allow this, for it may be a more self-directed learning approach. Nonetheless, self-directed learning is an important adult learning approach that shall prepare the dental student well for their lifelong career as a dentist (reference: Sandars, J. and Walsh, K., 2016. Self-directed learning. Education for Primary Care27(2), pp.151-152). Following live demonstrations, students may get the chance to repeat the procedure themselves (experiential learning), where the demonstrators can assess the students work, and provide constructive feedback. This can help dental students plan productively for their next learning experience, thus, enabling progress around the experiential (learning by doing) learning cycle. Such feedback also strongly supports the process of reflection, and enables learning of more advanced subject matter (reference: Modha, B., 2021. Experiential learning without prior vicarious learning: An insight from the primary dental care setting. Education for Primary Care32(1), pp.49-55). E-learning may not offer this practical opportunity. Also, live demonstrations may take place in a simulated dental environment, which is important for dental students’ familiarity and community of practice. However, e-learning may take place in non-dental environments (i.e., libraries, classrooms, students rooms, etc) (reference: Suksudaj, N., Lekkas, D., Kaidonis, J., Townsend, G.C. and Winning, T.A., 2015. Features of an effective operative dentistry learning environment: students' perceptions and relationship with performance. European Journal of Dental Education19(1), pp.53-62). Perhaps these points to include the benefits and limitations of live demonstrations, and e-learning, backed up by relevant references, need to be discussed within the manuscript.
  6. The manuscript delivers the pertinent topic of how dental education is changing, and that dental educational establishments need to provide digital learning approaches to improve the quality of dental education, especially in today’s pandemic era.
  7. The paper may be strengthened if the authors can consider: what makes their paper unique and stand out from the other similar ones that are already in publication?
  8. The paper is structured methodically and generally, the content flows well. The figures, tables and statistics are satisfactory. Overall, the paper is written well and is fairly easy to read. The utilised references are fine. Please kindly note, the language, grammar, punctuation, spelling and sentence structures within the current paper, all must be thoroughly assessed and polished throughout to ensure a succinct and coherent read. Please obtain the necessary scientific English language reading and editing assistance if need be, so that the paper has the potential to be read enjoyably by the international readership.
  9. There appears to be discrepancies in terms of capitalised words. In line 28, it would be expected that “college of dentistry” should be “College of Dentistry” as this is a name of an educational institute. In line 27, does “Dentistry” need to be capitalised? Please review this and ensure accurate consistency throughout.
  10. Please note, pre-clinical” as written throughout your paper, can be written as preclinical.
  11. Please pay attention to spelling. In line 68, faculties has been misspelt as “faculuties.”
  12. In line 84, “e-Learning” has been written, but elsewhere, “E-learning” has been written. Please ensure correct consistency throughout.
  13. What does ATLS stand for in line 98? Not all readers will know this, so please spell this out fully for the readership.
  14. This article is likely to be of interest to other dentists and allied dental professionals working within dental education establishments. This article may be of interest to healthcare professionals involved within teaching and mentoring lines of work.  
  15. Thank you for the opportunity to review this manuscript. I found it very interesting to read. It does have merit. If the aforesaid additions and alterations are made, then this paper shall have much more potential for publication.

Author Response

We thank the reviewer for his valuable inputs on our manuscript. All the comments raised by the reviewer are being addressed in point to point clarification file which is attached herewith.

Reviewer 3 Report

Dear authors, 

Congratulations for your work which is well written, respecting the requirements of a scientific paper.

Even if the number of participants is not big I found the study interesting.

The Introduction is well written. Please reconsider the last phrase of this section, containing the aim of the study, is too long and also please check the English Language.( lines 102-109). Also, in line 68 there is a spelling error.

In my opinion, Regarding the Materials and Methods, I would recommend to describe more clearly the measured parameters, as they appear in the lines 184-187, in the Results section. In the Discussion Chapter this parameters are also taken into considerations related to other articles in the scientific literature but they are not clearly presented in the Materials and Methods section, as I wrote above. Also, in the end of the Discussions I recommend to write more clearly about the limitations of this study and also about suggestions for future research.

Author Response

(The authors gave the same response as above.)

Round 2

Reviewer 1 Report

Title: Enhancement of skill competencies in Operative Dentistry using Procedure Specific Educational Videos (E-learning tools)  post COVID 19 era - A randomized controlled trial

Thank you for allowing me to revise the corrections and the re-edits in this manuscript. 

The authors have made a significant effort to improve the readability of this manuscript, and this is very clear. However, there are issues; some are minor, and one specific point is of significant concern, I believe.

Minor editing issues.

On lines 97 to 100, it reads, "Recently, Elham Soltanimehr discovered that virtual learning is better than traditional lecture-based learning for knowledge acquisition augmentation during the diagnostic imaging of bone lesions of the jaw."

Comments: The reference should be Soltanimehr et al. because this work is shared and not just by Elham Soltanimehr. Then it refers to that in lines 100 to 101, where it reads, "Therefore, he has suggested that virtual educational programs must 100 be revised to improve the student's reporting skills [20]."

Comments: This refers to Solanimher et al., so it should be, they suggested. This is also repeated on line 299, where it reads, "Recently, Elham Soltanimehr"

Lines 171 to 173 read, "At the end of the study, a questionnaire containing seven questions was circulated among the students to get their feedback to assess their perception about both the teaching methodologies and compare them."

Comments: You may need to address the validity and reliability of this questionnaire.

In lines 274 to 278, it reads, "However, contrary to this, some studies have found no difference in the competency level of the students whether they have been given video demonstration or traditional teaching [26]. Because the students have different psychomotor skill levels dividing them into other groups and evaluating the effect of the two methods might not necessarily show the real impact [26]."

Comments: It is unclear how the reference to this study relates to your study.

Major concerns:

The primary concern in this study is in the methodology and relates to the exclusion criteria. I have referred to this in the initial review.

It reads in the methodology, in lines 117 to 121, "Male and female second-year dental students of College of Dentistry, Jouf University, who have passed their prerequisite courses for the preclinical operative dentistry skill course were recruited in the study voluntarily after signing an informed consent form. The dental undergraduate students with any psychomotor disability were excluded from the study.

Firstly, the term psychomotor disability is unusual. I have searched for the use of this term in this context, and I did not find this term being used in this context.

Secondly, from the description, those who are described as with "psychomotor disability" have passed their prerequisite courses for the preclinical operative dentistry skill course, as described in these two sentences. If they passed, why were they excluded? In addition, if the assumption is that these are weaker students than the recruited in the study, wouldn't that be a source of bias?

I wrote previously, and the response in points 5 and 6 were not satisfactory.

My Comments Male and female second-year dental students of College of Dentistry, Jouf University, who have passed their prerequisite courses for the preclinical operative dentistry skill course, were recruited in the study voluntarily after signing an informed consent form. Comment: What is the study designed to investigate if the participants have already "passed their prerequisite courses for the preclinical operative dentistry skill course"? Please clarify.

Authors' response: We appreciate the concern of the reviewer regarding study design and its validity. To clarify this point we would like to inform that Preclinical operative skill course is taught to students in 2nd year of the Bachelor of Oral & Dental Surgery Program. For student to enroll in to this preclinical operative skill course the students need to successfully pass some of basic courses which are taught in First year. These basic courses which are taught in first year are called as prerequisite courses. The students gain the basic information such as anatomical and histological aspects of tooth and physiology of oral cavity from these prerequisite courses. This basic information is very much useful in understanding the concepts of cavity preparation and restoration in operative skills. We hope that now the concept of prerequisite course is clarified.

I do not believe the response clarifies my above comments.

Furthermore in point 6:

My Comments: The dental undergraduate students with any psychomotor disability were excluded from the study." Comment: The term psychomotor disability is an unusual term. If this term is to be accepted, how can there be students in dentistry categorized, according to the authors, as having a psychomotor disability? Please clarify.

Authors' response: It is appreciated that the reviewer is concerned about the exclusion criteria of the study and its validity. No doubt the dentistry is all about art and science with Psychomotor learning, development of organized patterns of muscular activities guided by signals from the environment. Behavioral examples include driving a car and eye-hand coordination tasks such as sewing, throwing a ball, typing, operating a lathe, and playing a trombone. Hence the authors in the current study also made sure that any participant lacking with psychomotor disability were excluded from the current investigation as the outcome variable might be influenced by the psychomotor disability. Hence an initial psychomotor disability screening was done for the control and experimental group using a simple test such as carefully looking at the participant's speech patterns, facial expressions, eye movements, posture, and body movements for signs of psychomotor slowing.

The response does not address the concerns. It does not clarify what does the term psychomotor disability means in this context. I have searched, and I could not find it in this context. In addition, this psychomotor disability screening is not clear and does not seem a reliable method or supported by any evidence? Furthermore, these are students who are within the school and to be screened and excluded by this described method (of questionable reliability) on the basis that they have a psychomotor disability; I am not sure they would be happy after this experience?

This is, to me is, the main concern in this manuscript. Therefore, according to what I understood from the description in the methodology, particularly the exclusion criteria and the response from the authors, I cannot but reject acceptance of this manuscript from.

Author Response

Respected Reviewer,

We thank you for your valuable comments.  All the comments of the reviewer are being addressed using point to point clarification process attached herewith.

Regards

Reviewer 2 Report

  1. There are still little inconsistencies with the language, grammar, punctuation, spelling and sentence structure. This all must be carefully evaluated and refined to ensure a concise and coherent readability. Please enlist the required scientific English language reading and editing assistance if necessary, to make certain that the paper holds the capacity to be read enjoyably by the international readership.

Some examples:

  • “doesn’t” on line 351. Such words do not embody scientific writing standards.
  • “Learning by educational videos doesn’t support the concept of directed self learning” – lines 350-351. This is an incorrect statement. Learning by educational videos (e.g., watching dental procedural videos) could well be one approach to self-directed learning. Please ensure that all statements within the paper are accurate and convey the right, intended meanings.
  1. Overall, the additions and alterations have improved the quality of this manuscript – it is now at a better standard to warrant publication. However, the above must be taken into account and fully addressed before publication is actually granted.

Author Response

(The authors gave the same response as above.)
